# BiDPO: Compositional Text-to-Image Generation Via Region-aware Bimodal Direct Preference Optimization

## Abstract

Despite the rapid progress of text-to-image (T2I) models, generating images that accurately reflect complex compositional prompts (covering attribute bindings, object relationships, counting) still remains challenging. To address this, we propose BiDPO, a framework to enhance T2I model's capability of compositional text-to-image generation. We begin by introducing an carefully designed pipeline to construct a large-scale preference dataset, BiComp, with strictly quality control. Then, we extend Diffusion DPO to jointly optimize image and text preferences, which is shown to greatly effective in improving the models to follow complex text prompt in generation. To further enhance the models for fine-grained alignment, we employ a region-level guidance method to focus on regions relevant to compositional concepts. Experimental results demonstrate that our BiDPO substantially improves compositional fidelity, consistently outperforming prior methods across multiple benchmarks. Our approach highlights the potential of preference-based fine-tuning for complex text-to-image tasks, offering a flexible and scalable alternative to existing techniques.

## 1 Introduction

Text-to-Image (T2I) generation has witnessed remarkable advancements in recent years, largely driven by the rapid development of diffusion models (Peebles & Xie, 2023; Esser et al., 2024; Betker et al., 2023; Labs, 2024). While existing models excel at generating images with high fidelity and aesthetics quality, they still struggle to accurately follow complex text instructions, especially when there are multiple objects, different attributes binding to each object, and complex inter-object relationships like spatial and numeracy involved (Huang et al., 2023).

To address these challenges, the research community has explored a variety of strategies. Some previous works introduce additional modalities, such as layouts (Zhang et al., 2024), scene graphs (Li et al., 2024b), or semantic panels (Feng et al., 2023) to provide structural guidance for the image generation process. While these approaches have achieved notable improvements, they heavily relies on supplementary inputs that may be difficult to obtain in practice. Another line of work seeks to enhance model comprehension through the integration of Large Language Models (Lian et al., 2023a) as a tool; however, such methods can be unstable and computationally intensive. **Motivated by this, we aim to enhance the compositional generation ability under pure text conditions**, without relying on external tools or modalities.

Direct Preference Optimization (DPO) (Rafailov et al., 2023), a powerful variant of Reinforcement Learning from Human Feedback (RLHF), refines traditional reward-model-based RLHF methods and has shown considerable promise in aligning generative models with human preferences. Despite its potential, the application of DPO to compositional text-to-image generation remains largely unexplored. We posit that DPO is particularly well-suited for this domain, as it can effectively leverage human feedback to enhance a model's ability to interpret and generate intricate compositions. Importantly, as a post-training technique, DPO can be applied to any pre-trained text-to-image model without requiring additional inputs or substantial architectural modifications, thereby offering a simple yet flexible and efficient solution.

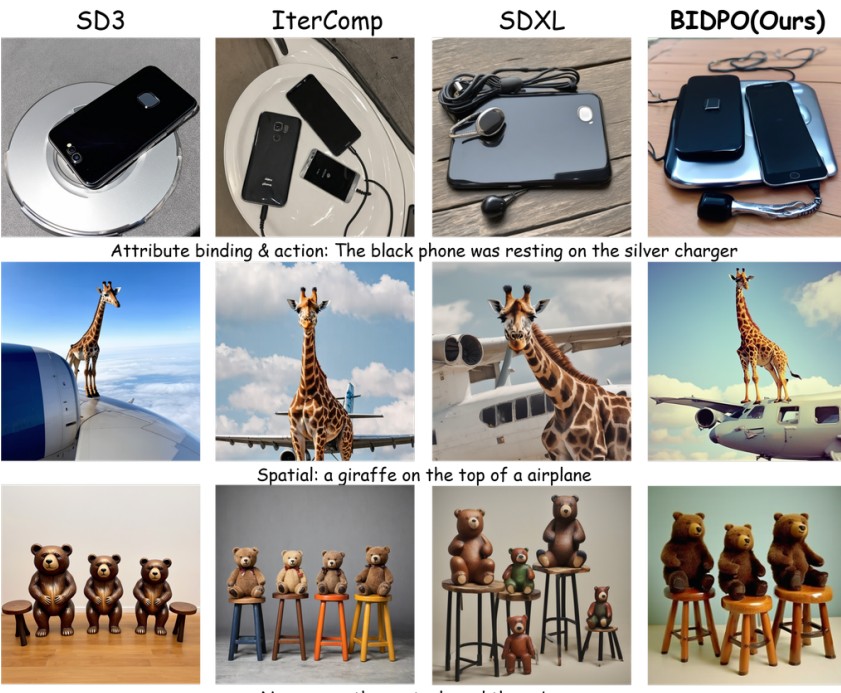

Figure 1: **Visualization of text-to-image generation results**. From left to right are Stable Diffusion 3 (Esser et al., 2024), IterComp (Zhang et al., 2025a), Stable Diffusion XL (Podell et al., 2023), and Stable Diffusion XL finetuned with our proposed BIDPO.

In this work, we introduce BIDPO, a novel framework that employs Bimodal Direct Preference Optimization to advance compositional text-to-image generation. Our approach is distinguished by a fully automated data pipeline for generating high-quality preference data, comprising the following stages: (1) collecting composition-related captions from diverse sources and generating corresponding images using a pre-trained text-to-image model; (2) regenerating captions for these images via a pipeline that integrates object detection, segmentation, and labeling; (3) editing the regenerated captions to produce distinct variants and utilizing an image editing model to modify the original images accordingly; and (4) applying a VQA-based filtering step to ensure the fidelity of the resulting image-caption pairs. The resulting dataset is characterized by high quality, diversity, large scale, and minimal visual differences between preference pairs—attributes essential for effective DPO training.

Subsequently, we extend Diffusion DPO (Rafailov et al., 2023) to a bimodal formulation that jointly considers image and text preferences, and employ this method to fine-tune a pre-trained Stable Diffusion model on the generated preference data. To further enhance model robustness and realism, we incorporate real-world data from the VisMin (Awal et al., 2024) dataset, thereby increasing the diversity and authenticity of the training corpus. Additionally, we introduce a region-aware training loss that accentuates specific regions of the image corresponding to edited captions. This, in conjunction with minimal visual differences in other regions, enables the model to more effectively learn and apply compositional modifications. Experimental results on T2I-CompBench (Huang et al., 2023) shows that our method leads to an average of 17% improvement in "attribute binding" category and a overall 10% improvement over the base model, demonstrating the effectiveness of our approach.

Our contributions are summarized as follows:

- We introduce BIDPO, a novel framework that improves model alignment by performing fine-grained preference optimization on both text and image modalities.

- We propose a region-level guidance mechanism that selectively steers the model's focus toward regions of interest. This mechanism is shown to substantially enhance the capability for fine-grained text-to-image alignment.

- We developed an automated data pipeline to construct a large-scale, high-quality text-to-image preference dataset, which includes both textual and visual negative examples. The proposed BᴵCᴏᴍᴘ comprise 57,474 original images and 94,502 edited images, covering six dimensions: color, shape, texture, spatial relationship, non-spatial relationship and numeracy.

- We conducted extensive experiments on several widely-used benchmarks, demonstrating significant performance gains over previous state-of-the-art methods.

## 2 RELATED WORKS

### 2.1 COMPOSITIONAL TEXT-TO-IMAGE GENERATION

The field of text-to-image (T2I) generation has undergone rapid progress with the emergence of large-scale diffusion models. These models are capable of synthesizing highly realistic images conditioned on textual prompts, and recent systems such as Stable Diffusion 3 (Esser et al., 2024), DALL-E 3 (Betker et al., 2023), and Flux (Labs, 2024) have achieved strong performance on standard quality benchmarks. Nevertheless, accurately capturing compositional semantics—involving multiple objects, attributes, and relations—remains a persistent challenge. Recent benchmark studies, including T2I-CompBench (Huang et al., 2023), GenEval (Ghosh et al., 2023) and DPG-Bench (Hu et al., 2024), highlight that state-of-the-art models often fail on fine-grained object binding and spatial reasoning tasks. Multiple methods have been proposed to address these limitations, such as incorporating structured scene representations (Feng et al., 2023; Zhang et al., 2024; Li et al., 2024b), conducting more precise control by generating the foreground objects and background separately (Xie et al., 2023; Lian et al., 2023a), leveraging large vision-language models to improve understanding (Lian et al., 2023a), employing contrastive learning techniques (Han et al., 2025), and introducing reinforcement learning strategies (Zhang et al., 2025a). Our work complements these approaches by focusing on preference-based optimization techniques to further align T2I models with human expectations on compositional tasks.

### 2.2 REINFORCEMENT LEARNING IN IMAGE SYNTHESIS

Preference alignment has become a central strategy for bridging the gap between model generations and human expectations. Early approaches adapt Reinforcement Learning from Human Feedback (RLHF), which is originally developed for large language models, to the image domain by training reward models or synthetic comparisons and optimizing with on-policy algorithms such as PPO (Lee et al., 2023; Xu et al., 2023). However, RLHF pipelines are computationally expensive and unstable when applied to high-dimensional image spaces. To address these limitations, Direct Preference Optimization (DPO) (Rafailov et al., 2023) was proposed as a simpler, more stable alternative that bypasses reinforcement learning by directly optimizing a contrastive preference objective. While DPO was first studied in language generation, recent works have begun adapting it to diffusion models, showing promising improvements in human preference alignment(Wallace et al., 2023). These results suggest that preference-based optimization without explicit reward modeling provides a practical pathway for fine-grained alignment in image synthesis. However, existing studies primarily focus on overall image quality and safety, with limited exploration of compositional capabilities. Our work extends the application of DPO to compositional T2I tasks, demonstrating that it can effectively enhance models' abilities to handle complex object interactions and attributes.

## 3 METHOD

### 3.1 PRELIMINARY

**Diffusion DPO.**

Diffusion DPO (Wallace et al., 2023) is a recent advancement in the field of diffusion models, which applies the principles of Direct Preference Optimization (DPO) to enhance the training of diffusion models. The core idea is to leverage human feedback in the form of preference data to guide the model towards generating outputs that are more aligned with human preferences. In Diffusion DPO,

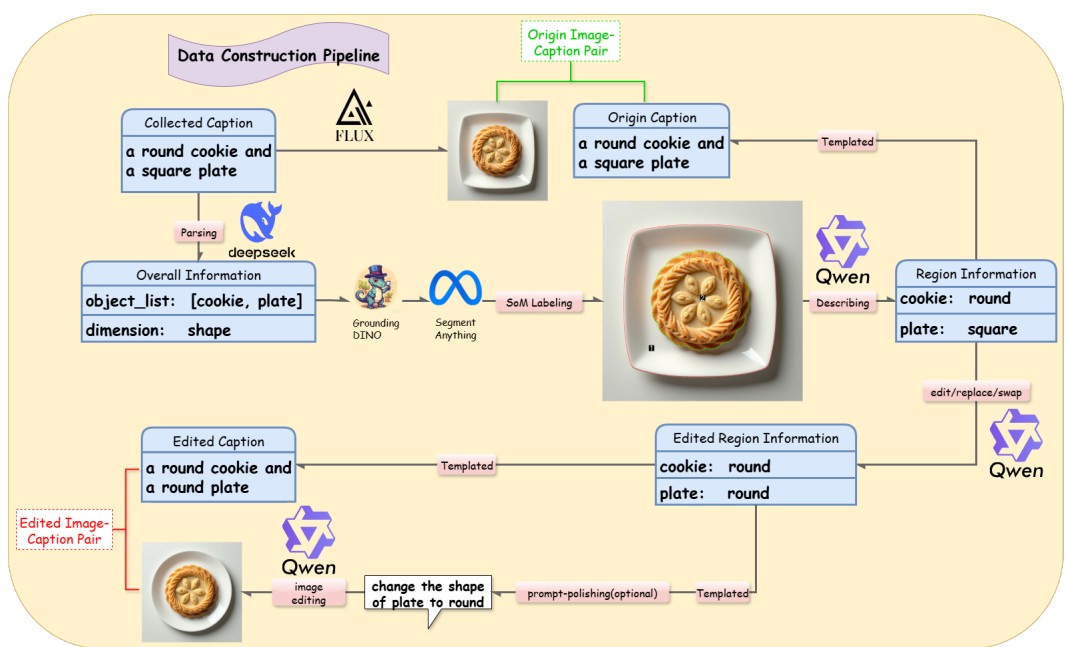

Figure 2: The data construction pipeline of our BICOMP dataset.

Table 1: **Number of images in each dimension.** Each original image may correspond to multiple edited images.

|  | Color | Shape | Texture | Spatial | Non-spatial | Numeracy | Total |
|---|---|---|---|---|---|---|---|
| Numer of Original Image | 19714 | 5399 | 9728 | 7919 | 3647 | 11067 | 57474 |
| Number of Edited Image | 46006 | 8473 | 17345 | 7919 | 3647 | 11112 | 94502 |

the training loss is defined as:

$$\mathcal{L}_{\text{DiffusionDPO}}(\theta) = - \mathbb{E}_{(\boldsymbol{x}_0^w, \boldsymbol{x}_0^l) \sim \mathcal{D}, \ t \sim \mathcal{U}(0,T), \ \boldsymbol{x}_t^w \sim q(\boldsymbol{x}_t^w | \boldsymbol{x}_0^w), \ \boldsymbol{x}_t^l \sim q(\boldsymbol{x}_t^l | \boldsymbol{x}_0^l)}$$
$$\log \sigma \left( -\beta T \omega(\lambda_t) \right) \Big($$
$$\|\boldsymbol{\epsilon}^w - \boldsymbol{\epsilon}_\theta(\boldsymbol{x}_t^w, t)\|_2^2 - \|\boldsymbol{\epsilon}^w - \boldsymbol{\epsilon}_{\text{ref}}(\boldsymbol{x}_t^w, t)\|_2^2$$
$$- \left( \|\boldsymbol{\epsilon}^l - \boldsymbol{\epsilon}_\theta(\boldsymbol{x}_t^l, t)\|_2^2 - \|\boldsymbol{\epsilon}^l - \boldsymbol{\epsilon}_{\text{ref}}(\boldsymbol{x}_t^l, t)\|_2^2 \right) \Big) \tag{1}$$

where $\mathcal{D}$ is the dataset of preference pairs, $\boldsymbol{x}_0^w$ and $\boldsymbol{x}_0^l$ are the preferred and less preferred samples respectively, $t$ is a randomly sampled time step, $q(\boldsymbol{x}_t | \boldsymbol{x}_0)$ is the forward diffusion process, $\boldsymbol{\epsilon}_\theta$ is the model's noise prediction, $\boldsymbol{\epsilon}_{\text{ref}}$ is the reference model's noise prediction, $\beta$ is a scaling factor, and $\omega(\lambda_t)$ is a weighting function based on the noise level at time step $t$.

## 3.2 DATA PIPELINE

**Prompt Collection and Image Generation.** We collect composition-related captions from various sources, including: CONPAIR (Han et al., 2025), ReasonGen-R1 (Zhang et al., 2025b), T2I-R1 (Jiang et al., 2025), T2I-CompBench Training Set (Huang et al., 2023).

For each collected caption, we generate 2-4 images using Flux.1-dev (Labs, 2024).

**Caption Generation.** Considering that the generated images may not always perfectly align with the original captions, we employ a caption generation pipeline to create new captions that better describe the generated images. The pipeline includes the following steps:

- **Dimension Parsing:** We use DeepSeek-V3 (DeepSeek-AI et al., 2024) to parse the original captions and identify which dimension the caption is referring to ("color", "shape", "texture",

"spatial", "action", "numeracy" or "other"). If the caption refer to multiple dimensions, we select one with the following priority (from highest to lowest): object relationship(spatial, action), numeracy, attribute binding(color, shape, texture). If the caption does not refer to any of the specified dimensions, we classify it as "other".

- **Object List Parsing:** We use DeepSeek-R1 (DeepSeek-AI et al., 2025) to extract the list of objects mentioned in the original captions.

- **Grounding Dino Detection and SAM Segmentation:** We use Grounding Dino (Liu et al., 2023) to detect objects in the generated images based on the object list extracted in the previous step. We then use SAM2 (Ravi et al., 2024) to segment the detected objects and obtain their masks.

- **VLM Describing:** We use Qwen2.5-VL-72B-Instruct (Bai et al., 2025) to label each segmented object in the image. First we label the image with SoM (Set-of-Mark) masks, which are highlighted regions in the image. Then, we ask Qwen to describe each masked object in detail, including its attributes (e.g., color, shape, texture) or relationships with other objects. We use specific prompts to guide the model based on the dimension identified in the first step.

- **Caption Synthesis:** Finally, we synthesize a new caption by combining the labels generated in the previous step. We use a template-based approach to ensure that the new caption is coherent and accurately describes the content of the image. In addition, for the "numeracy" dimension, we skip the VLM labeling step and directly use the result from Grounding Dino to count the number of objects and generate a caption accordingly.

We also filter out image-caption pairs that contain too many objects, with the consideration of the bad performance of detection and segmentation models in such cases and the convenience of the following image editing step.

**Caption Editing and Image Editing.** To generate the preference data, we first edit the regenerated captions to create distinct versions. We use Qwen2.5-VL-72B-Instruct (Bai et al., 2025) to generate distinct region information (attributes, relationships) based on the image with SoMs and the original region information. Then, we use Qwen-Image-Edit (Wu et al., 2025) model to edit the original image based on specific prompts. These prompts are designed to reflect the changes made in the edited captions, which are also generated in a template-based manner. For "action" and "numeracy" dimensions, Considering the complexity of editing images with multiple objects, we enhance the prompts by adding more detailed instructions using Qwen2.5-VL-72B-Instruct.

In order to enhance the model's ability to correctly attribute properties to objects, we add three more edited captions for each image-caption pair in the "color", "shape", and "texture" dimensions when the number of object is exactly two:

- Swap the attributes of the two objects. For example, if the original caption is "A red ball and a blue cube", the edited caption would be "A blue ball and a red cube".

- Replace the attributes of one object with the same attribute of another object. For example, if the original caption is "A red ball and a blue cube", the edited captions would be "A red ball and a red cube" and "A blue ball and a blue cube".

**Creatilayout Generation.** For the "spatial" dimension, it is hard to edit the image to reflect the changes in the edited caption. We use a different pipeline to generate the source and edited image-caption pairs. First, we use DeepSeek-V3 (DeepSeek-AI et al., 2024) to parse the original caption and generate a layout that describes the whole scene. Then, we use DeepSeek-V3 (DeepSeek-AI et al., 2025) again to edit the layout to a distinct version which differs in spatial relationships. Finally, we use CreatiLayout (Zhang et al., 2024) to generate images based on these layouts.

**VQA-based Filtering.** We employ a VQA-based filtering step to ensure the quality of the generated image-caption pairs. We use Qwen2.5-VL-72B-Instruct (Bai et al., 2025) to answer specific questions about the content of the images based on their captions. If the model's answers do not align with the expected responses, we discard those image-caption pairs. This step helps to ensure that the captions accurately describe the content of the images and that any edits made are reflected in both the images and their corresponding captions. The final dataset composition is shown in Table 1, and some samples are shown in Figure 3.

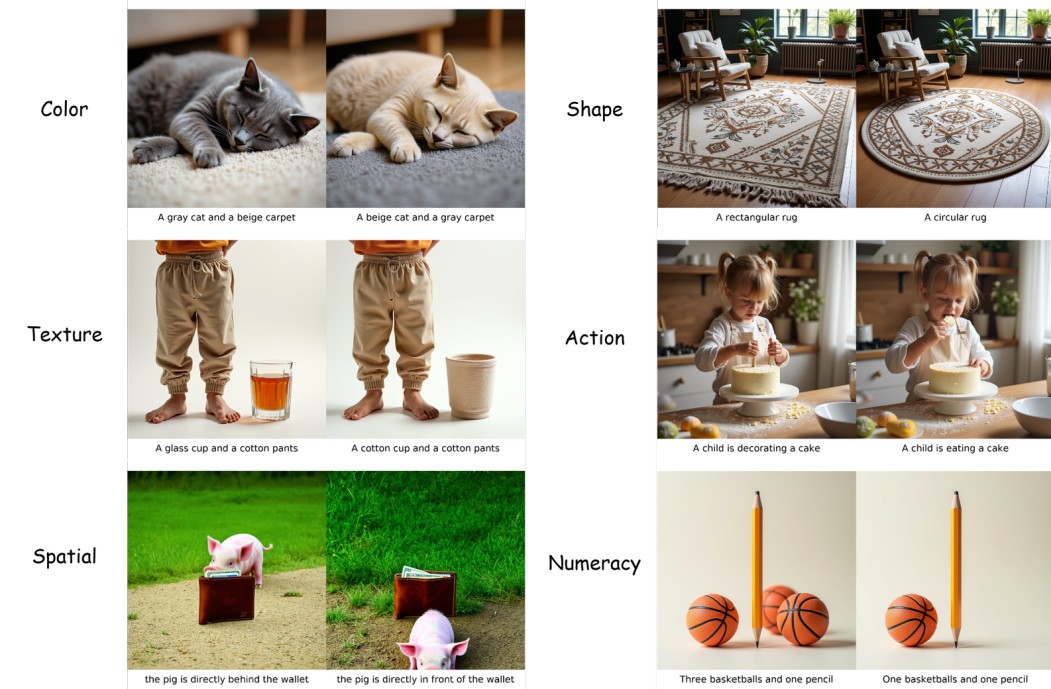

Figure 3: **Samples of each dimension in our BICOMP dataset.** For each group, the left image is generated from the original caption, and the right image is generated from the edited caption.

### 3.3 BIDPO

**Bimodal DPO.** We extend the Diffusion DPO to a text-based version that focuses on text preferences. The training loss is defined as:

$$
\begin{aligned}
\mathcal{L}_{\text{TextDPO}}(\theta) = &- \mathbb{E}_{(\boldsymbol{x}_0^w, \boldsymbol{y}^w, \boldsymbol{y}^l) \sim \mathcal{D}, \, t \sim \mathcal{U}(0,T), \, \boldsymbol{x}_t^w \sim q(\boldsymbol{x}_t^w | \boldsymbol{x}_0^w)} \\
&\log \sigma \left( -\beta T \omega(\lambda_t) \right) \big( \\
&\|\boldsymbol{\epsilon}^w - \boldsymbol{\epsilon}_\theta(\boldsymbol{x}_t^w, t, c^w)\|_2^2 - \|\boldsymbol{\epsilon}^w - \boldsymbol{\epsilon}_{\text{ref}}(\boldsymbol{x}_t^w, t, c^w)\|_2^2 \\
&- \big( \|\boldsymbol{\epsilon}^l - \boldsymbol{\epsilon}_\theta(\boldsymbol{x}_t^w, t, c^l)\|_2^2 - \|\boldsymbol{\epsilon}^l - \boldsymbol{\epsilon}_{\text{ref}}(\boldsymbol{x}_t^w, t, c^l)\|_2^2 \big) \big)
\end{aligned}
\tag{2}
$$

where $\mathcal{D}$ is the dataset of preference pairs, $\boldsymbol{x}_0^w$ is the preferred image, $\boldsymbol{y}^w$ and $\boldsymbol{y}^l$ are the preferred and less preferred captions respectively, $t$ is a randomly sampled time step, $q(\boldsymbol{x}_t | \boldsymbol{x}_0)$ is the forward diffusion process, $\boldsymbol{\epsilon}_\theta$ is the model's noise prediction which also conditioned on text embeddings $c^w$ and $c^l$, $\boldsymbol{\epsilon}_{\text{ref}}$ is the reference model's noise prediction, $\beta$ is a scaling factor, and $\omega(\lambda_t)$ is a weighting function based on the noise level at time step $t$.

If we look into the original Diffusion DPO loss, we can see that it basically depresses the diffusion process of the less preferred sample while enhancing the diffusion process of the preferred sample. In our TextDPO loss, we keep the same idea but change the less preferred sample to be the preferred image with the less preferred caption. However, this approach only considers the preference in one modality (text) while ignoring the other modality (image). We argue that both modalities should be considered to fully capture the preferences.

We consider adding image preferences in a implicit way, by using another preference data pair for training. For each image-caption pair $(\boldsymbol{x}_0^w, \boldsymbol{y}^w)$ and $(\boldsymbol{x}_0^l, \boldsymbol{y}^l)$, we create two training samples: one with $(\boldsymbol{x}_0^w, \boldsymbol{y}^w, \boldsymbol{y}^l)$ and another with $(\boldsymbol{x}_0^l, \boldsymbol{y}^l, \boldsymbol{y}^w)$. This way, the model learns to prefer the correct caption for each image while also considering the less preferred caption in the context of the other image. Concretely, through the TextDPO loss, the model learns to prefer caption $\boldsymbol{y}^w$ over $\boldsymbol{y}^l$ for image $\boldsymbol{x}_0^w$, which means that image $\boldsymbol{x}_0^w$ and caption $\boldsymbol{y}^l$ are the less preferred pair and this diffusion process should be depressed. Similarly, through the second training sample, the model learns to prefer caption $\boldsymbol{y}^l$ over $\boldsymbol{y}^w$ for image $\boldsymbol{x}_0^l$, which means that image $\boldsymbol{x}_0^l$ and caption $\boldsymbol{y}^l$ are the preferred

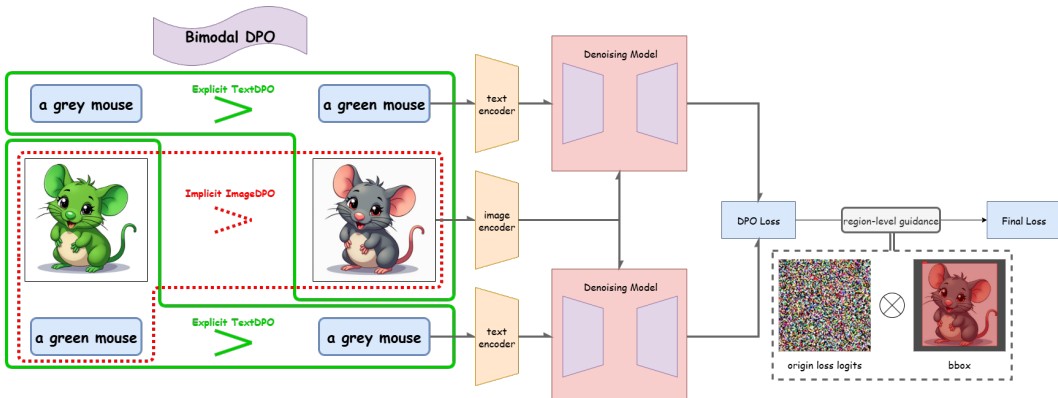

Figure 4: **Overview of our proposed BIDPO.** We conduct preference optimizing from both image and text side, and introduce a region-level guidance on the most related regions.

pair and this diffusion process should be enhanced. If we combine these two information together, we can see that the model implicitly learns to prefer image $x_0^l$ over $x_0^w$ for caption $y^l$. And also the same way, the model implicitly learns to prefer image $x_0^w$ over $x_0^l$ for caption $y^w$. In this way, the model learns to consider both image and text preferences during training.

**Region-level Guidance for Fine-grained Alignment.** To further enhance the model's ability to focus on specific regions of the image that correspond to the edited captions, we introduce a region-level guidance method. This method adjusts the importance of different regions in the image during training, helping the model to better understand and learn the desired modifications. We define the region-level guidance method as follows:

$$\mathcal{L}_{\text{BIDPO-region}}(\theta) = \mathcal{L}_{\text{BIDPO}}(\theta) \odot M \tag{3}$$

where $M$ is a mask that highlights the regions of the image corresponding to the edited captions, and the operator $\odot$ denotes element-wise multiplication. The mask is generated according to the bounding boxes of the objects involved in the edits, which are obtained from the caption generation and editing pipeline. We set a smaller weight for the regions not involved in the edits, ensuring that the loss is focused on the relevant regions of the image.

## 4 EXPERIMENTS

### 4.1 EXPERIMENTAL SETUPS

**Implementation Details.** We use Stable Diffusion XL (SDXL) (Podell et al., 2023) as our base model and fine-tune it with LoRA (Hu et al., 2022) and set rank to 8. We train the model for 200 steps with an effective batch size equals to 2048. The learning rate is set to 2048 * 4e-8 with a constant schedule and 50 warm-up steps. All experiments are conducted on $4\times$ H100 GPUs, with a total runtime of 13 hours. For the region-level guidance method, we set the weight to 1 for regions-of-interest and 0.5 for external regions to guide the model to focus on these regions. We do not use region-level guidance for data related to object numeracy or spatial relationships, as understanding these concepts requires a global focus. For the training data, we use 53k samples in total, combining 42k from our BICOMP dataset with 12k from VisMin (Awal et al., 2024) dataset.

**Evaluation Benchmarks.** We evaluate the effectiveness of our method on three challenging benchmarks designed to assess compositional capabilities in text-to-image generation, *i.e.* T2I-CompBench (Huang et al., 2023), GenEval Ghosh et al. (2023) and DPG-Bench Hu et al. (2024).

### 4.2 MAIN RESULTS

**T2I-CompBench.** T2I-Compbench (Huang et al., 2023) is a challenging benchmark that focuses on evaluating models in compositional generation, including object attributes and inter-object relationships. As shown in Table 2, our method achieves significant improvements over the baseline SDXL

Table 2: **Main Results on T2I-CompBench (Huang et al., 2023)**.

| Model | Attribute Binding | | | Object Relationship | |
|---|---|---|---|---|---|
| | Color | Shape | Texture | Spatial | Non-Spatial |
| Stable Diffusion 2 (Rombach et al., 2021) | 50.65 | 42.21 | 49.22 | 13.42 | 30.96 |
| GLIGEN (Li et al., 2023) | 42.88 | 39.98 | 39.04 | 26.32 | 30.36 |
| LMD+ (Lian et al., 2023a) | 48.14 | 48.65 | 56.99 | 25.37 | 28.28 |
| InstanceDiffusion (Wang et al., 2024b) | 54.33 | 44.72 | 52.93 | 27.91 | 29.47 |
| Attn-Exct v2 (Chefer et al., 2023) | 64.00 | 45.17 | 59.63 | 14.55 | 31.09 |
| PixArt-α (Chen et al., 2023a) | 68.86 | 55.82 | 70.44 | 20.82 | 31.79 |
| ECLIPSE (Patel et al., 2023) | 61.19 | 54.29 | 61.65 | 19.03 | 31.39 |
| Dimba-G (Fei et al., 2024) | 69.21 | 57.07 | 68.21 | 21.05 | 32.98 |
| GenTron (Chen et al., 2023b) | 76.74 | 57.00 | 71.50 | 20.98 | 32.02 |
| GORS (Huang et al., 2023) | 66.03 | 47.85 | 62.87 | 18.15 | 31.93 |
| ELLA (Hu et al., 2024) | 72.60 | 56.34 | 66.86 | 22.14 | 30.69 |
| MARS (He et al., 2024) | 69.13 | 54.31 | 71.23 | 19.24 | 32.10 |
| EVOGEN (Han et al., 2025) | 71.04 | 54.57 | 72.34 | 21.76 | 33.08 |
| SDXL (baseline) | 58.90 | 46.90 | 53.13 | 21.23 | 31.20 |
| SDXL-BIDPO | 79.35 ↑20.4 | 60.47 ↑13.6 | 71.36 ↑18.2 | 23.41 ↑2.2 | 32.29 ↑1.1 |

Table 3: **Main Results on GenEval (Ghosh et al., 2023)**.

| Model | Single Obj. | Two Obj. | Counting | Colors | Position | Color Attri. | Overall |
|---|---|---|---|---|---|---|---|
| SDv2.1 (Rombach et al., 2021) | 0.98 | 0.51 | 0.44 | 0.85 | 0.07 | 0.17 | 0.50 |
| PlayGroundv2.5 (Li et al., 2024a) | 0.98 | 0.77 | 0.52 | 0.84 | 0.11 | 0.17 | 0.56 |
| Show-o (Xie et al., 2024) | 0.95 | 0.52 | 0.49 | 0.82 | 0.11 | 0.28 | 0.53 |
| Emu3-Gen (Wang et al., 2024a) | 0.98 | 0.71 | 0.34 | 0.81 | 0.17 | 0.21 | 0.54 |
| IF-XL (Deep Floyd) | 0.97 | 0.74 | 0.66 | 0.81 | 0.13 | 0.35 | 0.61 |
| FLUX (Labs, 2024) | 0.98 | 0.81 | 0.74 | 0.79 | 0.22 | 0.45 | 0.66 |
| DALL-E 3 (Betker et al., 2023) | 0.96 | 0.87 | 0.47 | 0.83 | 0.43 | 0.45 | 0.67 |
| SDXL (baseline) | 0.95 | 0.68 | 0.42 | 0.85 | 0.11 | 0.19 | 0.53 |
| SDXL-BIDPO | 1.00 ↑0.05 | 0.86 ↑0.18 | 0.59 ↑0.17 | 0.88 ↑0.03 | 0.19 ↑0.08 | 0.22 ↑0.03 | 0.62 ↑0.09 |

model, especially in the attribute binding tasks (color, shape, texture). This demonstrates our method is effective in enhancing the model's ability to correctly associate attributes with their corresponding objects. Overall, our method achieves a substantial increase in the average score across all categories, highlighting its effectiveness for compositional text-to-image generation. Compared to other models designed for compositional generation, such as GLIGEN (Li et al., 2023), LMD+ (Lian et al., 2023b), and InstanceDiffusion (Wang et al., 2024b), our model still demonstrates a clear advantage. It worth noting that these models require an additional layout condition for control, whereas BIDPO achieves its strong performance using only the text prompts.

**GenEval**. We also evaluate our BIDPO on GenEval (Ghosh et al., 2023), a benchmark designed to assess text-to-image models in complex instruction following. As shown in Tab. 3, our BIDPO achieves clear improvements over the SDXL baseline model across most of the sub-tasks. The overall score shows a notable increase (0.62 vs. 0.53), which demonstrates our method's effectiveness in enhancing the base model to follow complex text prompts. Furthermore, our method even surpasses state-of-the-art models such as DALL-E 3 (Betker et al., 2023) and FLUX.1-dev (Labs, 2024) in several sub-tasks, including "single object" and "colors". This is particularly notable given our model is significantly smaller size and is trained on substantially less data.

**DPG-Bench** We also evaluate our method on DPG-Bench (Hu et al., 2024), a comprehensive benchmark for assessing the intricate semantic alignment capabilities of text-to-image models. As illustrated in Tab. 4, our BIDPO-SDXL achieves competitive results on the benchmark. Specifically, our model obtains comparable scores across all categories, including Global (83.92), Entity (85.28), Attribute (85.13), Relation (85.03), and Other (84.55), with a strong overall score of 78.84. Compared to the SDXL baseline (73.38 overall), our method demonstrates clear improvements, particularly in the Entity, Attribute, and Relation categories. These results validate the effectiveness and robustness of our approach for compositional text-to-image generation.

## 4.3 ABLATION STUDIES

We conduct extensive ablation studies to evaluate the key designs of BIDPO. We use SDXL as our baseline model, and explore several fine-tuning configurations:

Table 4: **Main Results on DPG-Bench (Hu et al., 2024).**

| Model | Global | Entity | Attribute | Relation | Other | **Overall** |
|---|---|---|---|---|---|---|
| PixArt- (Chen et al., 2023a) | 74.97 | 79.32 | 78.60 | 82.57 | 76.96 | 71.11 |
| PlayGroundv2 (Li et al.) | 83.61 | 79.91 | 82.67 | 80.62 | 81.22 | 74.54 |
| PlayGroundv2.5 (Li et al., 2024a) | 83.06 | 82.59 | 81.20 | 84.08 | 83.50 | 75.47 |
| Lumina-Next (Zhuo et al., 2024) | 82.82 | 88.65 | 86.44 | 80.53 | 81.82 | 74.63 |
| DALLE-3 (Betker et al., 2023) | 90.97 | 89.61 | 88.39 | 90.58 | 89.83 | 83.50 |
| SD3-medium (Esser et al., 2024) | 87.90 | 91.01 | 88.83 | 80.70 | 88.68 | 84.08 |
| SDXL (baseline) | 82.44 | 81.87 | 81.17 | 80.54 | 79.77 | 73.38 |
| SDXL-BɪDPO | 83.92 ↑1.5 | 85.28 ↑3.4 | 85.13 ↑4.0 | 85.03 ↑4.5 | 84.55 ↑4.8 | 78.84 ↑5.4 |

Table 5: **Ablation on key designs.** We report the overall scores over each benchmark.

| Method | T2I-CompBench | GenEval | DPG-Bench |
|---|---|---|---|
| SDXL | 43.57 | 53.29 | 79.86 |
| SDXL-SFT | 43.34 | 52.29 | 79.90 |
| SDXL-ImageDPO | 45.58 | 53.00 | 81.11 |
| SDXL-TextDPO | 13.48 | 4.71 | 39.39 |
| SDXL-BɪDPO w/o region-level guidance | 53.10 | 60.71 | 83.52 |
| SDXL-BɪDPO w/ region-level guidance | 54.37 | 62.14 | 83.79 |

- **SFT**: Supervised fine-tuning on our dataset without preference optimization.

- **ImageDPO**: Applying Direct Preference Optimization (DPO) using only image preferences (positive and negative images).

- **TextDPO**: Applying Direct Preference Optimization (DPO) using only text preferences (positive and negative texts).

- **BɪDPO** (w/o region-level guidance): Our method with bimodal DPO, using both positive and negative images and texts.

- **BɪDPO** (w/ region-level guidance): Combining bimodal DPO with region-level guidance based on bounding box annotations.

**Effectiveness of Bimodal Preference Optimizing.** As shown in Sec. 4.3, directly performing supervised fine-tuning on the composition-aware dataset fails to guide the model to focus on attribute binding and object relationships. In contrast, ImageDPO achieves a certain degree of performance improvement. This highlights the importance of guiding the model to focus on fine-grained compositional attributes through the comparison between positive and negative examples via direct preference optimization. However, solely perform text comparison leads to significant performance drop. In contrast, simultaneously optimizing preferences from both images and text more effectively promotes the model's cross-modal alignment, leading to a highly significant performance improvement.

**Effectiveness of Region-level Guidance.** From the last two lines of Tab. 5, it can be observed that the introduction of region-level guidance on top of BɪDPO leads to further improvements (1.2% on T2I-CompBench and 1.4% on GenEval). This indicates that explicitly guiding the model to focus on regions in the image that are relevant to the text description can effectively enhance the models to achieve fine-grained cross-modal alignment.

## 5 CONCLUSION

In this work, we present BIDPO, a novel method that introduces Direct Preference Optimization (DPO) to compositional text-to-image generation as well as extends it to a bimodal version and further enhances it with region-level scaling. Trained on our created compositionally-aware preference dataset, BIDPO significantly improves the compositional capabilities of text-to-image models, as demonstrated by extensive experiments on three standard benchmarks: T2I-CompBench, GenEval, DPG-Bench. For future work, we plan to extend our method to other kinds of generative models like autoregressive models, and more diverse generative tasks.

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

# 6 APPENDIX

In this appendix, we provide additional details and results as follows:

- In Section 6.1, we provide the full results of the ablation study on SDXL based models.
- In Section 6.2, we provide more details about our data construction process, including the composition of collected captions from various sources and the prompts used in various stages.
- In Section 6.3, we provide more visualization results of our BICOMP dataset and our BIDPO method.

## 6.1 ABLATION STUDY DETAILS.

The full results of the ablation study on SDXL based models are shown in Table 6, Table 7 and Table 8.

Table 6: **Ablation Study on T2I-CompBench (Huang et al., 2023).**

| Model | Attribute Binding | | | Object Relationship | | Numeracy |
|---|---|---|---|---|---|---|
| | Color | Shape | Texture | Spatial | Non-Spatial | |
| SDXL | 58.90 | 46.90 | 53.13 | 21.23 | 31.20 | 50.08 |
| SDXL-SFT | 58.67 | 46.65 | 52.21 | 21.13 | 31.28 | 50.08 |
| SDXL-ImageDPO | 67.39 | 53.12 | 59.42 | 23.4 | 30.82 | 39.34 |
| SDXL-TextDPO | 23.32 | 16.22 | 14.62 | 0.26 | 20.3 | 6.13 |
| SDXL-BIDPO w/o region-level guidance | 77.04 | 57.43 | 68.89 | 23.19 | 32.19 | 59.83 |
| SDXL-BIDPO w/ region-level guidance | 79.35 | 60.47 | 71.36 | 23.41 | 32.29 | 59.33 |

Table 7: **Ablation Study on GenEval Ghosh et al. (2023).**

| Model | Single Obj. | Two Obj. | Counting | Colors | Position | Color Attri. | **Overall** |
|---|---|---|---|---|---|---|---|
| SDXL | 0.95 | 0.68 | 0.42 | 0.85 | 0.11 | 0.19 | 0.53 |
| SDXL-SFT | 0.95 | 0.68 | 0.37 | 0.85 | 0.09 | 0.20 | 0.52 |
| SDXL-ImageDPO | 0.99 | 0.78 | 0.15 | 0.89 | 0.14 | 0.23 | 0.53 |
| SDXL-TextDPO | 0.13 | 0.01 | 0.01 | 0.11 | 0.01 | 0.02 | 0.04 |
| SDXL-BIDPO w/o region-level guidance | 1.00 | 0.83 | 0.52 | 0.90 | 0.16 | 0.23 | 0.61 |
| SDXL-BIDPO w/ region-level guidance | 1.00 | 0.87 | 0.56 | 0.90 | 0.17 | 0.23 | 0.62 |

Table 8: **Ablation Study on DPG-Bench Hu et al. (2024).**

| Model | Global | Entity | Attribute | Relation | Other | Overall |
|---|---|---|---|---|---|---|
| SDXL | 82.44 | 81.87 | 81.17 | 80.54 | 79.77 | 73.38 |
| SDXL-SFT | 82.68 | 81.94 | 79.52 | 81.02 | 81.03 | 73.23 |
| SDXL-ImageDPO | 79.13 | 83.19 | 82.76 | 83.39 | 82.49 | 75.70 |
| SDXL-TextDPO | 40.07 | 40.44 | 43.14 | 43.92 | 44.82 | 23.98 |
| SDXL-BIDPO w/o region-level guidance | 85.46 | 84.22 | 84.45 | 85.28 | 84.18 | 77.53 |
| SDXL-BIDPO w/ region-level guidance | 83.92 | 85.28 | 85.13 | 85.03 | 84.55 | 78.84 |

## 6.2 DATA CONSTRUCTION DETAILS

**Caption Collection.** Table 9 shows the number of captions collected from each source.

**LLM Prompt for Dimension Parsing.** Our dimension parsing prompt is shown in Listing 1. We use DeepSeek-V3 (DeepSeek-AI et al., 2024) as our LLM to parse the captions.

Listing 1: prompt for dimension parsing

```
# Task Description
Given a sentence, analyze its content and determine which of the
    ↪ following dimensions it primarily describes:
- color: describes colors (e.g., "red", "yellow", "dark blue")
```

| Dataset | Number of Captions |
|---|---|
| CONPAIR (Han et al., 2025) | 13,432 |
| ReasonGen-R1 (Zhang et al., 2025b) | 23,470 |
| T2I-R1 (Jiang et al., 2025) | 7,223 |
| T2I-CompBench Training Set (Huang et al., 2023) | 5,600 |
| **Total** | **49,725** |

Table 9: Number of composition-related captions collected from various sources.

```
- shape: describes geometric forms or outlines (e.g., "round", "
    ↪ triangular", "curved")
- texture: describes textures (e.g., "smooth", "rough") or materials (e.g
    ↪ ., "a plastic chair", "a glass window")
- spatial: describes spatial relationships or positions (e.g., "on the
    ↪ table", "next to", "inside", "beneath")
- non-spatial: describes actions/events without spatial focus (e.g., "
    ↪ chasing", "biting")
- numeracy: describes quantities or numbers (e.g., "three apples", "four
    ↪ ", "two")
- others: when none of the above categories apply

# Priority Rules
If multiple dimensions are present, select according to this priority:
1. spatial and non-spatial have highest priority (equal)
2. numeracy comes next
3. color, shape and texture have equal priority (lower than above)
4. others is always lowest priority

# Output Format
Provide your analysis in exact JSON format as shown below. Only include
    ↪ the JSON object in your response.

{{
    "dimension": "selected_dimension"
}}

# Examples
Input: "The cube is on the shelf"
Output: {{ "dimension": "spatial" }}

Input: "Five rough textured stones"
Output: {{ "dimension": "numeracy" }}

Input: "The soft yellow pillow"
Output: {{ "dimension": "color" }}

# Input
The input sentence is: {positive_caption}

# Output
For this sentence, the dimension is:
```

**LLM Prompt for Object List Parsing.** We use the prompt shown in Listing 2 to parse the object list from captions. We use DeepSeek-R1 (DeepSeek-AI et al., 2025) as our LLM to parse the captions.

Listing 2: prompt for object list parsing

```
You are an expert in parsing textual sentences. Given a text that
    ↪ describing an image, you task is to identify and extract the main
    ↪ entities in the image.

# Requirements
```

```
- You should only put the main entities that are visually visible in the
   ↪ image.
- Make sure the entities you identify are concrete objects, not abstract
   ↪ concepts; objects like 'living room' or 'wind' should not be
   ↪ identified.
- Make sure these entity objects can be detected by an object detector.
- Only output the entities themselves, without their adjectives or
   ↪ descriptions; for example, output 'dog' instead of 'white dog'.

# Output format
Orgainze the identified main objects in the scene into a json dict like
   ↪ this:
{{
    "object_list": ["object 1", "object 2", ...]
}}
# Input
For the sentence: {caption}, please identify the main visible objects.
```

**Image Describing Details.** Before we prompt the VLM to do the describing tasks, we restrict the image to follow the following rules: 1) with dimension "color", "shape", or "texture", the image should contain one or two objects 2) with dimension "spatial" or "non-spatial", the image should contain exactly two objects. 3) no repeated classes in the image; each object must belong to a unique class. We use specific prompts for different dimensions. Examples of "color" and "spatial" dimension are shown in Listing 3 and Listing 4. The prompts for "shape", "texture", and "non-spatial" dimensions are similar to the "color" and "spatial" ones, respectively. We use Qwen2.5-VL-72B-Instruct (Bai et al., 2025) as our VLM to describe the images.

Listing 3: prompt for VLM describing, with dimension "color"

```
# Task explanation
Given an image with clearly marked regions-of-interest (each region is
   ↪ indicated by a numerical ID and contour lines), please:
1. Identify all visible regions-of-interest by their numerical IDs
2. For each region, determine the predominant color of the object
   ↪ contained within it
3. Describe colors using standard web color names (e.g., "red", "
   ↪ forestgreen", "royalblue")
4. Handle uncertainty cases appropriately

# Output Requirements:
- Strict JSON format
- For unclear cases: use "unknown" as color value
- Sort results by region ID in ascending order

Output Example:
{
  "color_predictions": [
    {
      "region_id": 1,
      "color": "red"
    },
    {
      "region_id": 2,
      "color": "unknown"
    }
  ]
}

# Special Instructions:
- Ignore background colors outside marked regions
- Focus on the dominant colors
- IDs and contour lines are only for reference. DO NOT use them for color
   ↪  analysis
```

Listing 4: prompt for VLM describing, with dimension "spatial"

```
# Task explanation
Given an image with two clearly marked regions-of-interest (each region
    ↪ is indicated by a numerical ID and contour lines), please:
1. Identify the two regions-of-interest by their numerical IDs
2. Determine the precise spatial relationship between the two objects
    ↪ contained within the two regions-of-interest, where:
  - The reference object should be the visually more salient/dominant
      ↪ object (typically larger, more central, or more prominent in the
      ↪  scene)
  - The target object's position is described relative to the reference
      ↪ object
  - Use specific spatial descriptors (e.g., "on the right of", "above",
      ↪ "behind")
3. Handle uncertainty cases appropriately when spatial relationships
    ↪ cannot be clearly determined

# Output Requirements:
- Strict JSON format
- For unclear spatial relationship: use "unknown"
- Always describe the target object's position relative to the reference
    ↪ object

Output Example:
{
  "reference_object_id": 1,
  "target_object_id": 2,
  "spatial_prediction": "in front of",
  "notes": "object 2 is in front of object 1"
}

For unknown cases:
{
  "spatial_prediction": "unknown"
}

# Special Instructions:
- Do not use unclear descriptions like "next to", "beside", "near", "
    ↪ close to", etc
- IDs and contour lines are only for reference. DO NOT use them for
    ↪ spatial relationship analysis
- If neither object is clearly more salient, default to using the lower
    ↪ ID as reference
```

**VLM prompts for Region Information Differing.** We use specific prompt for each dimension to generate distinct region information. Examples of "color" and "spatial" dimension are shown in Listing 5 and Listing 6. The prompts for "shape", "texture", and "non-spatial" dimensions are similar to the "color" and "spatial" ones, respectively. We use Qwen2.5-VL-72B-Instruct (Bai et al., 2025) as our VLM to generate the distinct region information.

Listing 5: prompt for VLM differentiation, with dimension "color"

```
# Task Explanation
Here is an image with outlined regions (each region is indicated by a
    ↪ numerical ID and contour lines).
And here are the list of regions with their dominant colors for reference
    ↪  (format: {{"region_id": N, "color": "color_name"}}):
{obj}

Now please propose a visually distinct color for each region that
    ↪ significantly differs from ALL provided dominant colors in the
    ↪ image.

# Requirements:
```

```
972   1. For each region, suggest ONE color that contrasts distinctly with ALL
973      ↪ dominant colors in the image
974   2. Consider human perceptual difference (avoid suggesting similar hues/
975      ↪ brightness)
976   3. Prefer standard color names (e.g., "red", "green")
      4. Never suggest the same as any input dominant color
977   5. When multiple options exist, choose the highest-contrast alternative
978
979   # Output Format (strict JSON):
980   {{
        "output": [
981       {{"region_id": N, "different_color": "color_name"}},
982       ...(other regions)
983     ]
984   }}
985
986   # Examples:
      Input Colors: [{{"region_id": 1, "dominant_color": "red"}}, {{"region_id
987      ↪ ": 2, "dominant_color": "blue"}}]
988   Output: {{
        "output": [
989       {{"region_id": 1, "different_color": "yellow"}},
990       {{"region_id": 2, "different_color": "black"}}
991     ]
992   }}
993
994   Input Colors: [{{"region_id": 1, "dominant_color": "green"}}, {{"
995      ↪ region_id": 2, "dominant_color": "yellow"}}]
      Output: {{
996     "output": [
997       {{"region_id": 1, "different_color": "magenta"}},
998       {{"region_id": 2, "different_color": "navy_blue"}}
999     ]
1000  }}
```

Listing 6: prompt for VLM differentiation, with dimension "spatial"

```
1003  # Task Explanation
1004  Here is an image with TWO outlined regions (each region is indicated by a
1005     ↪  numerical ID and contour lines).
      And here is the spatial relationship between the two objects contained in
1006     ↪  the two regions:
1007  object {object_id_1} is {spatial_relation} object {object_id_2}
1008
1009  Now please propose a geometrically distinct spatial relationship that
1010     ↪ significantly differs from the given relationship.
1011
1012  # Requirements:
      1. Suggest ONE primary spatial relationship that contrasts maximally with
1013     ↪  the input relationship
1014  2. Consider these transformation axes for differentiation:
1015     a) Vertical inversion (above/below     swap)
         b) Horizontal inversion (left/right     swap)
1016     c) Dimensional shift (adjacent     separated)
1017     d) Topological change (inside     outside)
1018  3. Use standard spatial terms from this vocabulary:
1019     [above, below, on the left of, on the right of, in front of, behind,
1020        ↪ ...]
      4. The new relationship must be:
1021     a) Physically plausible for the objects' shapes/sizes
1022     b) Perceptually distinct from original
         c) Expressed as "object X [RELATION] object Y"
1023  5. Include brief reasoning in "notes"
1024
1025  # Output Format (strict JSON):
```

```
{{
  "output": {{
    "different_spatial_relation": "relation_term",
    "notes": "object [object_id_X] [RELATION] object [object_id_Y]"
  }}
}}

# Examples:
Input: "object A is above object B"
Output: {{
  "output": {{
    "different_spatial_relation": "below",
    "notes": "object A is below object B (vertical inversion)"
  }}
}}

Input: "object X is inside object Y"
Output: {{
  "output": {{
    "different_spatial_relation": "outside",
    "notes": "object X is outside object Y (topological complement)"
  }}
}}

Input: "object 1 is adjacent to object 2"
Output: {{
  "output": {{
    "different_spatial_relation": "separated",
    "notes": "object 1 is separated from object 2 (proximity reversal)"
  }}
}}
```

**VLM prompt for VQA-based filtering.** We use the prompt shown in Listing 7 to filter out low-quality samples. We use Qwen2.5-VL-72B-Instruct (Bai et al., 2025) as our VLM to perform the filtering.

Listing 7: prompt for VLM VQA-based filtering

```
You are given an image with several regions of interest (ROIs). Each ROI
    ↪ is highlighted in the image with contour lines and labeled with a
    ↪ unique numerical ID.

You are also given a list of questions. Each question refers to one or
    ↪ more ROIs. Here are the questions:
{questions}

Your task:

1. For each question, evaluate whether the statement is correct with
    ↪ respect to the corresponding region(s).
2. Provide a confidence score between 0 and 1 ('answer') indicating how
    ↪ strongly you agree with the statement (1 = completely true, 0 =
    ↪ completely false).
3. Provide a short explanation ('reason') describing why you assigned
    ↪ this score.

The output format must strictly follow this JSON structure:

```json
[
  {{
    "question_id": <int>,
    "answer": <float between 0 and 1>,
    "reason": "<string explanation>"
  }},
```

```
  ...
]
```

**Example:**
Input image: contains region 1 (a yellow lemon) and region 2 (a red apple
  ↪ ).
Questions:

```json
[
  {{"question_id": 0, "question": "Does region 1 mark a yellow lemon?"}},
  {{"question_id": 1, "question": "Does region 2 mark a blue apple?"}}
]
```

Expected output:

```json
[
  {{"question_id": 0, "answer": 0.99, "reason": "Region 1 does mark a
    ↪ yellow lemon."}},
  {{"question_id": 1, "answer": 0.01, "reason": "The apple in region 2 is
    ↪  actually red."}}
]
```

## 6.3 MORE VISUALIZATION RESULTS.

We provide more visualization results of our BICOMP dataset and our BIDPO method in Figure 6
and Figure 5, respectively.

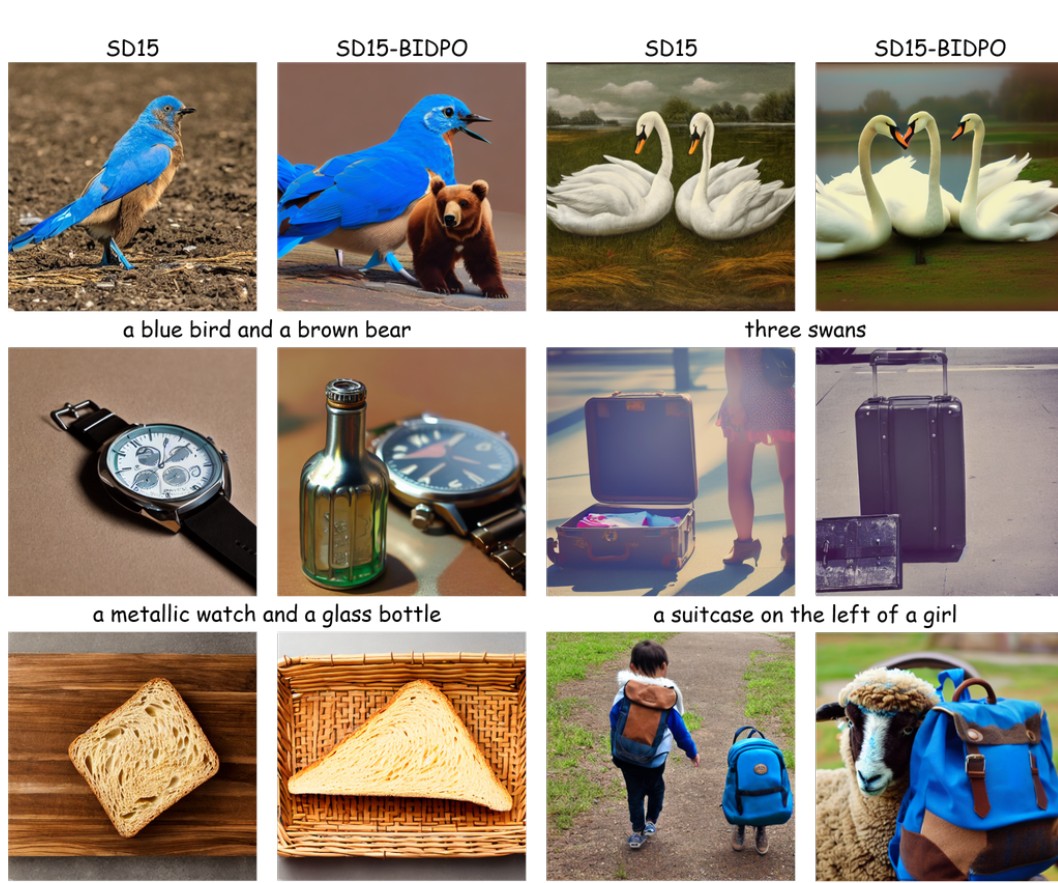

Figure 5: Visualization of text-to-image generation results of Stable Diffusion 1.5 finetuned with our proposed BıDPO, compared with the original Stable Diffusion 1.5.

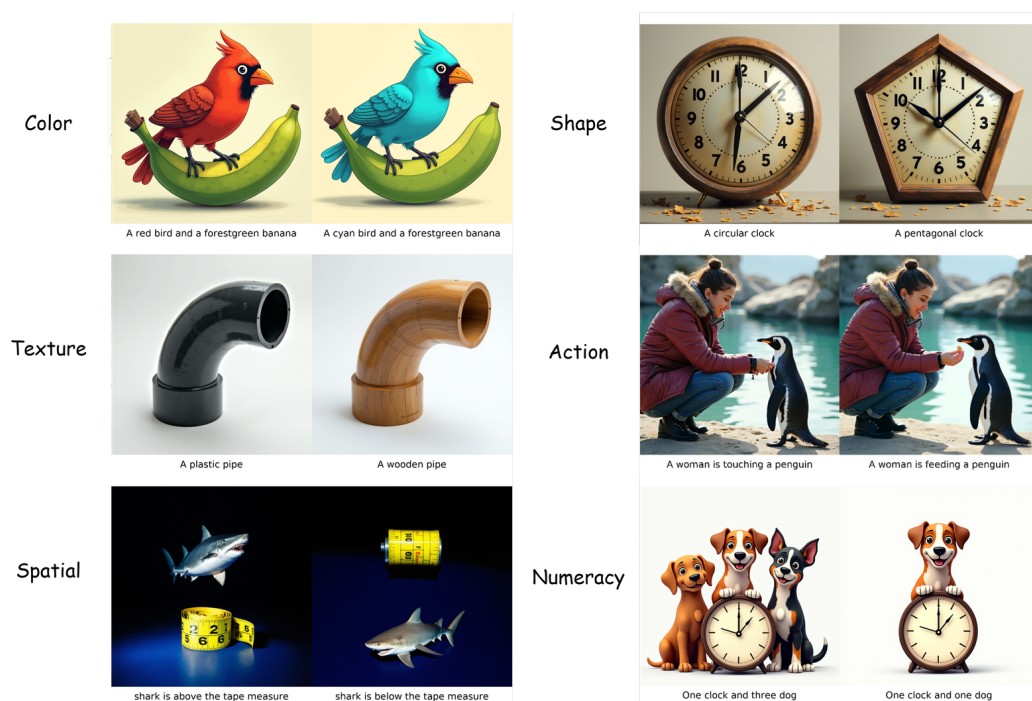

Figure 6: Samples of each dimension in our constructed preference dataset. For each group, the left image is generated from the original caption, and the right image is generated from the edited caption.

