# OpenReview forum: "Compositional Text-to-Image Generation Via Region-aware Bimodal Direct Preference Optimization"
_ICLR.cc/2026/Conference — ICLR 2026 Conference Withdrawn Submission_

### Official Review · Reviewer_SZEN · 2025-10-28

**Soundness:** 3
**Presentation:** 3
**Contribution:** 2
**Rating:** 4
**Confidence:** 4

**Summary:**

This paper presents BIDPO, a framework that extends Diffusion DPO to bimodal preference optimization for improving compositional text-to-image generation. The main contributions are jointly optimizing both image and text preferences, introducing region-level guidance to focus on relevant image regions, and developing an automated pipeline to construct a large-scale preference dataset.

**Strengths:**

1. Comprehensive data pipeline: The automated data construction process is systematic and well-designed, incorporating multiple quality control stages and featuring refined design specifically tailored for compositional text tasks.

2. Experiments show significant improvements over SDXL, with performance gains across multiple benchmarks.

**Weaknesses:**

1. Why does SDXL-SFT show performance degradation? Does this indicate that the quality of the constructed dataset is insufficient? Typically, diffusion-DPO also performs SFT before DPO, and SFT usually leads to performance improvements. Can you explain why SFT results in performance decline in this paper?

2. Although the metrics show significant improvements on benchmarks, visible distortions can be observed in the generated images in Figure 5. Can you design methods to enhance compositional text capabilities while maintaining perceptually acceptable image quality?

3. Can you explain the reasons for the collapse observed in TextDPO?

**Questions:**

Address the issues mentioned in the weaknesses.

---

### Official Review · Reviewer_LDgq · 2025-10-30

**Soundness:** 2
**Presentation:** 2
**Contribution:** 1
**Rating:** 2
**Confidence:** 5

**Summary:**

his work proposes BIDPO, a novel framework that improves model alignment by performing fine-grained preference optimization on both text and image modalities. In the standard DPO, the optimization process learns to prefer winning images over losing images for a given prompt. In this updated formulation, the BIDPO, proposes to also align the preference on correct prompt for a given image, where the incorrect prompt is obtained by slight perturbation and obtained a rigorous image editing/VLM verification framework. Additionally, it proposes a region-level guidance mechanism that guides the model’s focus towards regions of interest in the preference optimization objective. This mechanism is shown to substantially enhance the capability for fine-grained text-to-image alignment. Finally, the paper proposes BICOMP, a new preference dataset that consists of 57,474 original images and 94,502 edited images, covering six dimensions: color, shape, texture, spatial relationship, non-spatial relationship and numeracy. Preference optimization experiments on SDXL architecture with the proposed BIDPO methods show the efficacy of the proposed scheme.

**Strengths:**

- BICOMP dataset construction scheme looks interesting and would be helpful in other directions (including joint signal in image and text alignment) since there exists instructions on few edits and corresponding edited images.
- BIDPO seems novel in the addition of winning and losing text pairs in the DPO formulation. Although the text component in the BIDPO has not been flushed out and verified thoroughly on other models.

**Weaknesses:**

- There seems to be no comparison against existing popular DPO methods such as MAPO, SPO, RankDPO, Flow-GRPO, etc.
- The proposed method has been only applied on a single diffusion model SDXL. It is unclear if the scheme would translate to any other diffusion model, specially models which jointly processing image and text modalities.
- The region-level guidance seems to only provide marginal improvements compared to other components and does not provide any significant contribution to the paper.
- Many relevant works (see missing refs) have been omitted in the manuscript

**Questions:**

- For data creation, how do you ensure that the your generation pipeline rules out anatomically bad or incorrect / blurry generations from the Image generation models?
- It’s unclear how the proposed method compares against popular DPO methods such as Flow-GRPO, RankDPO, SPO, LPO, MAPO, etc.
- How does the method scale to models which jointly process image and text modalities such as SD3.5, Flux, Qwen-Image, etc.? Since at the center of this work is the fact that underlying diffusion model needs to understand the difference between correct and incorrect captions, it would be better to apply such an approach to a model which jointly processes image and text information.
- There seems to be some discrepancy between Tab 4 and Tab 5. The DPG-Bench score for SDXL (baseline) do not align. Tab 4 reports 73.38 and Tab 5 reports 79.86.
- SDXL-BIDPO with and without region-level guidance does not seem to be helping much on DPG-Bench and this is the only evidence of the region-level guidance approach.
- How does one extend the proposed method to video modality?


Missing References:
- Margin-aware Preference Optimization for aligning diffusion models without reference (MAPO): https://arxiv.org/abs/2406.06424
- SPO : Step-aware Preference Optimization: Aligning Preference with Denoising Performance at Each Step https://arxiv.org/pdf/2406.04314v1
- Flow-GRPO— Training Flow Matching Models via Online RL : https://arxiv.org/pdf/2505.05470
- RankDPO — Scalable Ranked Preference Optimization for Text-to-Image Generation:  https://arxiv.org/abs/2410.18013
- Latent Preference Optimization (LPO) — Diffusion Model as a Noise-Aware Latent Reward Model for Step-Level Preference Optimization   : https://arxiv.org/pdf/2502.01051
- DSPO — Direct Score Preference Optimization for Diffusion Model Alignment https://openreview.net/forum?id=xyfb9HHvMe
- Bridging SFT and DPO for Diffusion Model Alignment with Self-Sampling Preference Optimization https://arxiv.org/abs/2410.05255

---

### Official Review · Reviewer_PsCF · 2025-11-01

**Soundness:** 2
**Presentation:** 2
**Contribution:** 2
**Rating:** 2
**Confidence:** 3

**Summary:**

This paper introduces BiDPO, a novel framework to enhance compositional text-to-image generation by combining bimodal direct preference optimization (DPO) with region-aware guidance. The authors construct a large-scale, high-quality preference dataset (BiComp) using a carefully designed automated pipeline and extend Diffusion DPO to jointly optimize image and text preferences. Experimental results on T2I-CompBench, GenEval, and DPG-Bench demonstrate improvements over baseline in attribute binding and object relationships.

**Strengths:**

1. This paper introduces bimodal DPO, a novel framework to improve compositional T2I without the need for additional inputs or reliance on external tools.
2. It proposes a BiDPO formulation that jointly optimizes image and text preferences, which is a well-motivated extension of DPO methods.
3. The BiComp dataset is large and carefully constructed with strict quality control.
4. The method achieves consistent improvements across multiple benchmarks over its baseline.

**Weaknesses:**

1. The data construction pipeline may be potentially redundant. It uses multiple external tools (GroundingDINO, SAM etc.), which is computationally expensive and complex. As far as I know, Qwen2.5-VL alone may be capable of both detection and captioning,
What is the difference of using Qwen2.5-VL to write the region information and template-based new caption and your pipeline? Qwen-Image-Edit also allows image editing using text prompts, without the need for object masks.
2. Although the proposed method requires no additional inputs or integration of external LLM, DPO relies on a large, high-quality preference dataset which is expensive to create and may be sensitive to the quality and distribution of the data.
2. The paper lacks comparisons with alternative methods, such as Attend-and-Excite[A], RPG-DiffusionMaster[B], implemented on the same base model, making it difficult to assess relative performance gains.
4. Figure 4 is not explained, especially how TextDPO and ImageDPO are combined in practice.
5. The choice of state-of-the-art models for comparison could be more up-to-date  (e.g., Nano-banana[C], GPT-Image-1[D]) .

[A] Chefer, Hila, et al. "Attend-and-excite: Attention-based semantic guidance for text-to-image diffusion models." ACM transactions on Graphics (TOG) 42.4 (2023): 1-10.

[B] Yang, Ling, et al. "Mastering text-to-image diffusion: Recaptioning, planning, and generating with multimodal llms." Forty-first International Conference on Machine Learning. 2024.

[C] Google. Introducing gemini 2.5 flash image, our state-of-the-art image model. 2025.

[D] OpenAI. Gpt-image-1. 2025.

**Questions:**

1. Can you provide a more detailed explanation of how the bimodal DPO loss combines image and text preferences (e.g., in Figure 4)?
2. Why was LoRA chosen over full fine-tuning? Were any experiments conducted to compare their effectiveness?
3. Can you compare BiDPO with other compositional-generation methods (e.g., Attend-and-Excite) under the same base model?

---

### Official Review · Reviewer_qSVV · 2025-11-01

**Soundness:** 4
**Presentation:** 3
**Contribution:** 2
**Rating:** 4
**Confidence:** 4

**Summary:**

The paper presents an improved formulation for Direct Preference Optimization in diffusion modles. Speciifcally, the paper shows how images from altered prompts can be used to construct improved preference pairs (i.e. pairing the right caption for the image vs mismatched captions). In addition ot this, there's also a mask applied at the pixel/latent level to ensure that important regions in the sample are highlighted in the optimization. Results on prompt following benchmarks such as T2I-Compbench, GenEval, and DPG-Bench indicate that the method is able to bring notable improvements to the base SDXL model.

**Strengths:**

The paper proposes a very interesting and compelling idea of a) preparing preference pairs in a targeted manner with precise modifications between the positive negatives and b) an improved loss fomrulation where mismatched captions can be used for additional supervision.

The paper also achieves promising results on SDXL and is reasonably well-presented.

**Weaknesses:**

[Major]

Following Diffusion-DPO, there's been a bunch of work incorportaitng additional structure both in terms of the data construction and the loss into the formulation [a,b,c]. DenseDPO for instance applies the loss at finer temporal granularity for video models, which is quite close to the idea of having a spatial mask in images. Similarly, CaPO and RankDPO (among many other works in this direction) also demonstrate good improvements over the original Diffusion-DPO formulation, including on these same benchmarks. It would be a good idea for the paper to acknowledge these.

Additionally, an important caveat is regarding the source of data/prompts. A lot of previous DPO works (Diffusion-DPO/KTO, CaPO) rely on either Pick-a-Pic or DiffusionDB which have images that are quite different from T2I-Compbench/GenEval. Here, the data directly includes prompts from T2I-Compbench training set among other sources. While this is clearly not overfitting, it also might explain why the performance improvements on T2I-Compbench are so outsized compared to the gains on other benchmarks (which are still good, but not as dramatic).



[Minor]

I'd also be curious to see how the BiDPO formulation performs beyond the SDXL LoRA fine-tuning setting with newer models (e.g. with SD3/Flux etc.)

The effect of the region-guidance seems quite minimal, I would be curious if this is really necessary, or is there some scenario where it can be particularly useful?


[a] Lee et al. "Calibrated Multi-Preference Optimization for Aligning Diffusion Models", CVPR 2025

[b] Karthik et al. "Scalable Ranked Preference Optimization for Text-to-Image Generation", ICCV 2025

[c] Wu et al. "DenseDPO: Fine-Grained Temporal Preference Optimization for Video Diffusion Models", NeurIPS 2025

**Questions:**

I think broadly the paper introduces an interesting concept for improving text-to-image models, and I'm inclined towards accepting it. I'd mostly want clarity from the authors on a) the effect of the data source and b) contextualizing the paper better in the contex to existing works on diffusion model alignment (e.g. there are 25 papers here on DPO https://github.com/xie-lab-ml/awesome-alignment-of-diffusion-models). While it's impossible to exhaustively cover all the works, there must atleast be some effort to summarize the existing work in the field.

---

### Note · Authors · 2025-11-12

I have read and agree with the venue's withdrawal policy on behalf of myself and my co-authors.